# Demographic Resilience in the Rural Area of Romania. A Statistical-Territorial Approach of the Last Hundred Years

Ionel Muntele [1,2] , Marinela Istrate [1,*], Raluca Ioana Horea-Șerban [1] and Alexandru Banica [1,2]

1   Faculty of Geography and Geology, Alexandru Ioan Cuza University of Iasi, 700505 Iași, Romania;
    imuntele@yahoo.fr (I.M.); ralusel@yahoo.com (R.I.H.-Ș.); alexandrubanica@yahoo.com (A.B.)
2   Center of Geographical Studies, Romanian Academy, 700505 Iași, Romania
*   Correspondence: marinelaistrate75@yahoo.com

**Abstract:** The deep demographic crisis that Romania has been going through, like most Eastern European states, requires complex analyses. The present study aims at analyzing the numerical evolution of the rural population of Romania, extended for more than 100 years (1912–2020), on the basis of which a multivariate analysis was subsequently performed. The manifestation of specific patterns of territorial evolution and the identification of the incidence of some physical–geographical and socio-economic factors are investigated by means of the dynamics of certain distinct processes, such as rural depopulation or ability to adapt to various transitions. The identification of the fragile and dynamic areas allows discussions on the need for public policies better oriented towards mending excesses, illustrated by the persistence of some gaps, in terms of access to basic services. At the same time, the identification of trends towards a shrinking population system, with an increasingly evident concentration of the population in peri-urban areas or along major transport routes, calls for specific policies geared more towards environmental management in depopulated and depopulating areas, or towards creating the conditions for sustainable spatial planning in high-density areas. For the time being, the disadvantaged areas are rather ignored, while the extension of metropolitan areas is rather chaotic and poorly regulated.

**Keywords:** risk of depopulation; rural sustainability; population dynamics; driving forces; Romania

## 1. Introduction

The demographic crisis that Romania has been going through, like most Eastern European states, manifests itself as a combination of a negative natural balance and labor emigration. Affecting cities and villages alike, the current crisis is a challenge for the future of this country. The strong territorial disparities that are emerging, ranging from depopulation at local or regional level to the concentration of the population in metropolitan areas, create the premises for an inequitable development, especially to the detriment of rural areas. The extreme processes mentioned above can be considered natural from the perspective of demographic resilience and sustainable rural development. Reconfiguring the distribution of the population according to its specific development potential, adapted to the requirements of the contemporary society, can lead to the recalibration of population contingents, expressed either by contraction or by expansion. However, the risk of the depopulation of rural areas entails land management costs that are as significant, comparatively speaking, as those induced by peri-urban concentration. At the extreme, depopulation could have an impact on entire rural regions, threatening the disintegration of the settlement structure.

Although demographic decline could be accompanied by a real progress of productivity, especially after Romania's European integration, the capacity to support a viable economic system is seriously diminished in the affected areas. After 1990, this decline has been no longer offset by the growth of the urban population, the Romanian demographic

system being completely affected, revealing the manifestation of a deep crisis, at national level (Table 1).

**Table 1.** Numerical evolution and distribution of the population, by residential areas (according to the official administrative statute), in Romania (1912–2020). Source: our own elaboration on INS data.

| | Population (Thousands of Inhabitants) | | | | | Population Density (Inhabitants per km$^2$) | | | | |
|---|---|---|---|---|---|---|---|---|---|---|
| | 1912 | 1948 | 1966 | 1992 | 2020 | 1912 | 1948 | 1966 | 1992 | 2020 |
| Urban | 3436 | 4964 | 7388 | 12,829 | 10,456 | 112 | 161 | 240 | 417 | 340 |
| Rural | 9351 | 10,926 | 11,712 | 10,315 | 8872 | 45 | 53 | 56 | 50 | 43 |
| Total | 12,787 | 15,890 | 19,100 | 23,144 | 19,328 | 54 | 66 | 80 | 97 | 81 |

The factors that involve territorial divergent trends, in similar geographical contexts, have an uneven manifestation in time and space. A broader retrospective analysis is required in order to identify the mechanisms that generate these trends. For these reasons, the present study proposes an analysis of the evolution of the population dynamics in the rural areas of Romania over a period of time extending between 1912 and 2020. These rural areas include all 2861 officially recognized territorial administrative units although, from a purely demographic perspective, many small towns are experiencing similar trends. The typology identified after performing this analysis is doubled by a series of regressions that have a factorial database as a starting point, for each of the three historical periods overlapping the analyzed time interval: the period of the two world wars, the period marked by the communist regime and the period of adaptation to the market economy. The typological and factorial analyses are complementary, the first one trying to differentiate specific areas of manifestation of some evolution trends, the second one explaining, based on the processed information and applied model, the incidence of some natural or socio-economic factors. The information used was processed according to official sources and reconstituted in accordance with the current administrative structure.

## 2. Literature Review and Hypotheses Development

Resilience is a concept which is strongly inserted in the new geography current, becoming indispensable to the analysis of the way in which territorial systems and agencies respond to the stimuli of change [1]. Resilience summarizes the ability of the system to counteract the effects of exposure, disturbance or variability of transformative events [2]. Territorial demographic structures represent a key component in the analysis of the way in which socio-economic processes get adapted. Their resilience, less investigated, is most often integrated into the widely accepted notion of socio-ecological resilience [3]. Closely related to the concept of vulnerability, resilience is apparently its opposite, a vulnerable socio-ecological system being a system that has lost its resilience [4]. However, in order to become resilient, a system manifests certain vulnerability; consequently, the more it is exposed to disruptions, the more it practices its resilience capacity. Thus, in some recent works, demographic resilience is seen as the ability to maintain a long-term expansion trend (numerical growth) [5]. The most palpable and accessible indirect indicator is consequently the change in population size, expressed by the positive growth rate, which can prove resistance to regressive trends or recovery after a period of disruption. It also indicates a certain attractiveness, demonstrates adaptation and preservation of structures, functions and feedbacks and the promotion of future development options [6]. Therefore, small-scale demographic growth illustrates the resilience performance of a local system [7]. From this point of view, we agree with the conclusions of some well-known authors, who recommend resistance and recovery as the measurable components of resilience [8,9].

On the other end of the spectrum, any negative trend can be regarded as an expression of the risk of depopulation. Its impact on environmental sustainability, economic growth or social well-being is strong, being strictly associated with the rigidity of time-invariant territorial characteristics [10]. The analysis of the decline of the rural population, an

emerging field of research, is increasingly developing under the form of cross-sectional studies, applied to specific geographical areas, with an emphasis on its ecological impact and connections with sustainable development [11]. The topic is gaining ground in those countries which are largely affected by this process, such as Spain or China, while the mechanisms that generate it are becoming better known, thanks to studies using long-time data series [12]. The cited authors defined rural depopulation as "a complex interplay of chronic net out-migration and natural decrease", later on also adding the effects introduced by demographic transition (declining fertility and increasing mortality due to population aging). Although adapted to the North American context, this vision is close to the current situation of the rural areas in Eastern Europe, particularly the Romanian ones. As a result of the permanent redistribution of the population, accentuated by its modern tendency to concentrate in large urban agglomerations, the risk of depopulation emerged later in the east of the continent, against the background of a predominantly agrarian economy, closely linked to the evolution of the demographic transition. While in the west of the continent, it was felt beginning with the nineteenth century, in its south it fully manifested itself immediately after 1950 and in the east only after 1970, with a significant strengthening during the transition period [13] (pp. 16–23). Some authors regard the population decline induced by the risk of depopulation only as a strictly demographic problem, which can be solved through the recovery of the fertility indicators [14]. However, it is a much more complex issue, being closely linked to the globalization circuits that strengthen the center-periphery disparities [15]. The social and economic effects of depopulation, beyond any controversy, have a strong manifestation in the peripheral areas of developed countries as a consequence of the weakened productive capacity, brought about by workforce reduction and diminished innovation potential [16]. Enormous disparities, especially in terms of living standards, economic dependence and mass emigration, offer an even less optimistic outlook for large rural areas in Eastern Europe [17,18]. In this context, the link between the risk of depopulation and the resilience of the socio-economic structures becomes important, manifesting itself through the ability of the affected areas to adapt to permanent changes by maintaining an acceptable level of life quality [19]. In this respect, there are studies analyzing shrinking cities by using resilience theory as a filter to explain specific processes of shrinkage in relationship to certain "natural" phases of the adaptive cycles [20]. The decrease of the rural population below the level that can be supported by the territorial system should not necessarily be seen as a harmful effect of the population's tendency to concentrate in metropolitan areas. There are studies that do not indicate a direct correlation between population growth and economic growth [21]. The demographic dividend manifests itself only in the presence of a strong educational dividend [22]. From this point of view, Romania is in an unfavorable position, as indicated by the PISA tests and by the strong disparities between the rural and the urban environment [23,24]. The difficult access to education can emphasize the negative effects of demographic decline and fuel depopulation trends, as some studies suggest [25]. Formal education in rural areas suffers from much stronger gaps than in other countries, depopulation strengthening the lack of accessibility to primary education. Retired people choose to live in the countryside especially when it comes to peri-urban areas endowed with urban-type infrastructure. The isolated rural space does not benefit from attractive conditions in this respect.

Another major issue related to the depopulation of rural areas is the loss of traditional identities. Romania preserved its rural matrix for a longer period compared to Western countries, and unlike in other similar countries the Romanian villages consolidated during the communist period through systematization plans, which maintained the population but implied the cost of coercive measures that largely cancelled the traditional individuality of the peasant [26]. This identity loss continued in recent times, although the driving factors changed.

In the particular case of Romania, the first decade after 1990 witnessed a particular phenomenon, namely the retreat of the population towards rural areas, stimulated by the restitution of agricultural property. The decline of the urban population occurred

as a counter-urbanization process triggered by deindustrialization due to an inability to adapt to transition [27], the primary sector thus becoming a survival solution for many of those who had lost their urban jobs. Mainly occurring in plain or hilly areas with a surplus of agricultural land, this phenomenon has not been judiciously exploited by the authorities, the gaps between the rural and the urban environment growing deeper, while the urgency of developing a modern infrastructure and of diversifying the job offer was ignored. The impulse that could have revitalized the rural spaces already touched by depopulation was missed, this population flow being subsequently driven into a massive international migration [28], which disadvantaged peripheral and isolated mountainous areas [29]. During the nineties emigrants were originating mostly from urban areas and from the wealthier (western) regions of Romania. After 2002 the eastern (less developed) part of the country becomes the major region of origin for the Romanian emigration, and the population originating from rural areas becomes increasingly connected with various streams of emigration [30]. The international migration of the Romanian rural population was circular for a long time but, during the last decades, especially after Romania's accession to the EU (in 2007), it has become permanent. As a future perspective, we can expect the return of some of the emigrants (especially those who reach retirement age), but most of them are likely to settle down in peri-urban areas, where they can enjoy a lifestyle similar to the one they used to experience abroad.

There is a certain consistency in neglecting rural areas, the poverty of the Romanian village being endemic, an expression of the incapacity of the state to transform peasant agriculture into a productive and diversified economy [31]. In this context, the study of the risk of depopulation becomes a support for regional development policies, still insufficiently consolidated in Romania.

According to some authors, the factors that explain demographic vulnerability/lack of resilience and the risk of depopulation can be summarized in three key axes: poor transport infrastructure, environmental restrictions and difficulty in purchasing goods and services [32]. Similarly, other authors [33] consider that economic status, the level of service to the population and accessibility are the most important drivers of depopulation, as estimated by means of population dynamics. Colantoni et al. [5] (2020) also emphasize the importance of the basic geographical gradients, generated by the specific position (mountain–plain, urban–rural etc.), in connection with the use of long series of data on the numerical evolution of the population. Such studies also attest to a strong spatial self-correlation, suggesting that policies focused on creating local government clusters should take precedence in order to prevent or limit the depopulation of certain areas. The integration of some geographical factors such as relief, climate change or soil quality is considered desirable, even if it is difficult to build a theoretical model that includes them [34,35]. Nevertheless, changes in land-use intensity that are implied by these indicators demonstrate shifts towards new urban-rural relationships [36,37]. In order to estimate the advance of the depopulation process, an important indicator is the agricultural land abandonment, which has generated a vast literature [38]. The risk of contraction of the settlement network, of its disintegration, is largely dependent on depopulation, correlated with the abandonment of agricultural land use [39]. In Romania, the information on this phenomenon is incomplete and its extent can be detected rather indirectly by following the evolution of cultivated areas, without knowing if the decline observed in recent decades is due to land abandonment or change of use. The development of tourism, where possible, is sometimes a sustainable alternative, stimulating the preservation of a certain part of the population, especially in mountainous areas [40]. However, the cases in which the rural population has stabilized due to tourism development are rather isolated, at least until now.

Sustainable rural development can be regarded as an integrative concept which plays an important role in optimizing the negative trends induced by the risk of depopulation and in stimulating demographic resilience. Although in a continuous re-evaluation, this concept has developed in opposition to two alternative paradigms: the agro-industrial and the post-productivist models [41] (pp. 1–21). Marsden reformulates it as a "territorially based development that redefines nature, re-emphasizes food production and agro-ecology and re-

asserts the socio-environment role of agriculture as a major agent in sustaining rural economies and cultures" [41]. The present study, mainly focused on the analysis of demographic resilience and depopulation risk, touches only tangentially on the issue of sustainable rural development, which could represent a natural continuation of the current approach.

As some synthesis studies point out, rural decline is an inevitable phenomenon, whose intensity depends on the interactions between the rural environment and the external factors and whose challenge can be stopped by its resilience capacity when local actors work together towards rural revitalization [42]. Beyond its relentless nature, rural decline is dependent on the local context, creating a diversity of situations that require a multi-scale analysis [43].

Previous studies on Romania have revealed a west–east gradient in the manifestation of processes with a strong impact from the perspective of demographic resilience, such as the completion of the demographic transition, industrialization and planned urbanization during the communist period and emergence of the risk of depopulation [44]. They have also pointed out specific disparities that separate the mountainous areas from the hilly or plain ones, as well as some contradictions connected to the urban–rural relations or to the cultural context.

The analysis of the evolution of the main demographical components, reconstructed at national level for the 20th century [44], proves the conformity of the Romanian society to the classic demographic transition pattern, quite faithfully following its distinct stages: beginning, expansion and ending. It can be assessed that the modernization of the society starting with the second half of the 19th century brought about an increase in the fertility rate and a substantial drop in the death rate during the following decades, thus generating a significant population surplus, typical of the beginning of the demographic transition. The expansion stage can be placed between 1920 and 1960, with the intermezzo imposed by the Second World War. Despite the high values of the fertility rate, the precarious socio-economic situation, especially in rural areas, had a visible impact on infant mortality, which recorded the highest values in Europe (perhaps except for Albania), being more than double in comparison to the European average. The final stage of the demographic transition in Romania can be placed between 1960 and 1990, interrupted by the effects of the natalist policy promoted by the communist authorities (especially during 1967–1972 and 1983–1989). After 1990, we can speak about the installation of a post-transition regime in completely unfavorable economic and social circumstances. The signs of the new demographic regime ("the second demographic revolution" [45]) are visible in recent years, the variations of the two main demographic components (birth and death rates) generating a natural balance close to 0. Overcoming this phase of natural decline (which started in 1992) is going to be difficult in the context of the massive emigration of the young population, a demographic segment which is vital to the preservation of a fertility rate high enough to soften the forecasted dramatic decline in population over the coming decades.

As for the rural areas, they experienced a brutal intervention in the traditional production system by collectivization of agriculture, which generated a rapid population exodus during the communist period. This context favored the diffusion of the demographic transition even in the more conservative areas, consequently manifesting the depopulation process. After 1990, gradually, the internal rural exodus was replaced by an international exodus which amplified the mentioned process.

Starting from these theoretical premises, the present study proposes an ample and detailed chrono-spatial analysis of the evolution of Romania's population, in order to illustrate demographic resilience in Romania and its driving forces for a longer period of time. In order to fulfill it, two main objectives were formulated:

**Objective 1 (O1).** *To assess demographic resilience for a longer period with a focus on rural areas. The demographic resilience of the local and regional systems can be illustrated by the numerical evolution of the population, using a simple methodology, providing a sufficiently conclusive basis to frame sustainable rural development strategies. The correction of some observed deficiencies requires such an analysis in order to identify the areas marked by specific trends.*

**Objective 2 (O2).** *To evaluate the driving forces of demographic resilience. The irregular manifestation, in time and space, of the factors that can explain the evolution trends of the population is a certainty in the Romanian case as well, with features deriving from the specific socio-economic and political context. The incidence of physico-geographical characteristics can play an important role in directing processes such as depopulation.*

## 3. Materials and Methods

The information needed for the present study was provided by official statistics (censuses, databases of the National Institute of Statistics [46]), reports and research projects of some European or national institutions or it was extracted from cartographic materials.

The proposed working methodology comprises two types of analyses:

(a) A descriptive analysis of the evolution of the annual population growth rate, performed by means of an agglomerative hierarchical clustering (AHC), using the Xlstat program, developed by Addinsoft, 2015 version. The purpose of this operation was to obtain a typology of the evolution of the average annual population growth in order to outline the territorial differentiations and specific profile. The study area is represented by the 2861 rural communes in Romania, according to the last administrative-territorial division in force. As a result, it was necessary to reconstruct the numerical evolution by allocating the population of each village to the current commune. The database created includes 10 series expressed as a percentage, obtained by reporting the difference between two recordings to the average population of the reference range. The information reflects the results of the 10 official censuses carried out in Romania between 1912 and 2011, plus the resident population as estimated at the beginning of 2020 by the INS. For the regions that did not belong to the present territory of Romania in 1912 we used the data provided by the Austro-Hungarian census of 31 December 1910, taken as a reference by the census conducted in 1930 at the level of the entire present territory [47]. We refer to population censuses, except for the last period (2020), for which there are only official estimates of the resident population [46]. The population was reconstituted according to the current administrative division [48,49]. It is true that certain censuses (such as the ones conducted in 1941 and 1948) were carried out in difficult circumstances, which may have generated possible errors that we assume. However, the generally expressed trends show that the information is largely reliable.

(b) A set of multivariate analyses for each of the three separate periods (1912–1948, 1948–1992, 1992–2020). The dependent variable is the average annual growth rate, calculated for each of the periods. The explanatory variables differ in number from one period to another, depending on the available information (9 for 1912–1948, 11 for 1948–1992 and 16 for 1992–2020) [46–49]. The explanatory variables were standardized in relation to the maximum and minimum values, excluding outliers in the case of the variables expressed by numerical values and by granting factor scores, from 1 to 0, in the case of those expressed qualitatively (Table 2). For two of the variables (altitude and aging index) the values were hierarchically reversed, the lowest value receiving a Z score of 1. The standardized data were also operated in the Xlstat program, opting for a linear regression. In order to test the manifestation of some differences at the regional level in terms of the incidence of some factors, the same analyses were performed at the level of the three major cultural-historical divisions of Romania (Moldavia, Wallachia and Transylvania-Banat) or by the three major forms of relief (mountains, hills and plains).

**Table 2.** The standardization methodology of the variables used in the multivariate analysis.

| Variable | Data Source | Explanations | Standardization |
|---|---|---|---|
| AGR | CENSUS of 1912, 1948, 1992; Tempo Online-Database of INS | Average annual growth rate (%) | Z score |
| DMC | Road Atlas of Romania, 1:200,000 [50] | Distance from large and medium-sized cities (km) | Z score |
| ATI | Topographic map of Romania, 1:50,000 | Access to transport infrastructure: Railways and European roads | 1 |
| | | European roads only | 0.9 |
| | | Railways and National roads | 0.7 |
| | | National roads only | 0.5 |
| | | Railways and county roads | 0.3 |
| | | County roads only | 0.1 |
| | | Local roads only | 0 |
| ALT | Topographic map of Romania, 1:50,000 | Average altitude (m); 0–100 m were considered together, equivalent with the Z score 1. | Z score |
| DRG | Special report of European Court of Auditors (33/2018) [51] | Risk of drought: high/very high | 0.33 |
| | | Moderate | 0.66 |
| | | Low/absent | 1 |
| AFR | INS, Tempo-Online Database | Degree of afforestation: low (less than 10% of total surface) | |
| | | Medium | 0.66 |
| | | High | 0.33 |
| | | Very high | 0.05 |
| SQ | Map of soil types potential [52] | Soil quality: low | 0.33 |
| | | Average | 0.66 |
| | | Good | 1 |
| HF | Census of 1912–2011 | Habitat fragmentation (inhabitants/village) | Z score |
| DNS | Census of 1912–2011 | Population density (inhabitants/km$^2$) | Z score |
| NB | INS, Tempo-Online Database (for 1966–2019); 1930 Census. | Natural balance (‰); for 1912–1948 it was replaced with the share of 0–14 years. | Z score |
| AG | Census of 1930, 1966, 1977, 1992, 2002, 2011 | Aging index (+65/0–14 years) | Z score |
| PST | Census of 1966, 1977, 1992, 2002, 2011 | Share of the population employed in secondary and tertiary sector (% of active population) | Z score |
| MB | INS, Tempo-Online Database (for 1966–2019) | Migratory balance (‰) | Z score |
| HSE | Census of 2011 | Share of the population with higher and secondary education | Z score |
| INC | INS, Tempo-Online Database, Census of 2002 | Average income (lei/capita) | Z score |
| NBH | INS, Tempo-Online Database (for 1990–2019) | Newly built homes (related to total population) | Z score |
| SWC | Census of 2011 | Access to sewerage, water supply and central heating (related to total households) | Z score |

The selection of the variables for the multivariate analyses was dependent on the access to information but it also took into account the analysis methodology proposed by papers on similar topics. Duquenne and Hadjou [53] (2010) advanced some methods for the analysis of fragile rural areas, starting from the premise of the existence of three dimensions that can sensitize the evolution of the population: difficulties of coordination of actors, less valorization of resources and objective disabilities. The indicators used largely coincide with those proposed in the model above. More recently, Colantoni et al. [5] (2020) used, at the level of the whole Greece, a model with similar variables, adapted to its specific situation, primarily the presence of maritime coasts. In the case of Romania, we considered it necessary to introduce variables meant to test the importance of agricultural activities, including from the perspective of soil quality or drought risk, the Romanian rural area being still agricultural by excellence. The introduction of certain variables related to life quality (access to basic services) or which indicate the dynamics of the habitat (construction of new homes) is also related to the specificities of Romanian villages, still deficient in terms of development.

## 4. Results and Discussion

The results of the analyses were illustrated cartographically by using Adobe Illustrator CS12 or in tabular format. In the case of the typological analysis, a map was drawn up in order to reflect the distribution of the evolution types, specific evolution profile and specific dynamics at the levels of the years 1912 and 1992, respectively. The results of the linear regressions are illustrated by the correlative matrices, the R2 determination coefficient and the *p*-value correlation coefficient.

### 4.1. Typology of the Numerical Evolution of the Population during the Period 1912–2020

The agglomerative hierarchical clustering used identified the presence of six types (classes) with a distinct evolution profile. The quality of the classification, which used the Euclidean distance and the Ward aggregation method, is illustrated by the within-class variance of 0.3607, which has values well below the between-class variance of 0.6393.

The six classes create some coherent territorial structures, evidence of the strong regional differences in the evolution of the population dynamics induced either by the manifestation of certain gaps in the dynamics of the demographic transition process or by the character of the migration balance. The analysis of the evolution profiles shows that each class has a specific position in relation to the two key concepts presented: demographic resilience and depopulation risk (Figure 1). The first four classes have an approximately equal representation (between 341 and 369 communes), the last two being much more representative (855 and 572 communes, respectively), grouping half of the 2861 analyzed administrative units.

Class 1 can be considered as illustrative of demographic resilience, grouping especially communes situated in the vicinity of some important urban centers, rarely including more isolated communes, marked by a conservative demographic behavior. This periurban position provided them with a greater capacity of adaptation, both during the strong manifestation of the rural exodus (1956–1992) and during the transition period. The predominantly positive growth rate (except for the period 1977–1992), with very high values in recent decades, shows an attractiveness that can be explained by the tendency of exurbanization of many services or industrial activities; this is mainly the advantage of those communes that enjoy a favorable position in relation to transport infrastructure. Meanwhile, these communes lost their rural character and became highly dependent on the nearby city. As a consequence, some problems related to housing estates on the outskirts of large cities including infrastructure issues could also imply, in future, even the transformation of some of these areas in ghettoes. These problems mainly arise when the increase in population is the result of one wave of development which creates abnormal one-sided demographic and social structures. One can notice the remarkable extension of some periurban areas around the large urban centers (Bucharest, Iasi, Constanta, Timisoara,

Brasov, Cluj etc.), closely related to their economic dynamism. At the same time, however, important cities, such as the Galați-Brăila group, Craiova or Ploiești, did not generate a similar evolution, possibly correlated with their proximity to the capital.

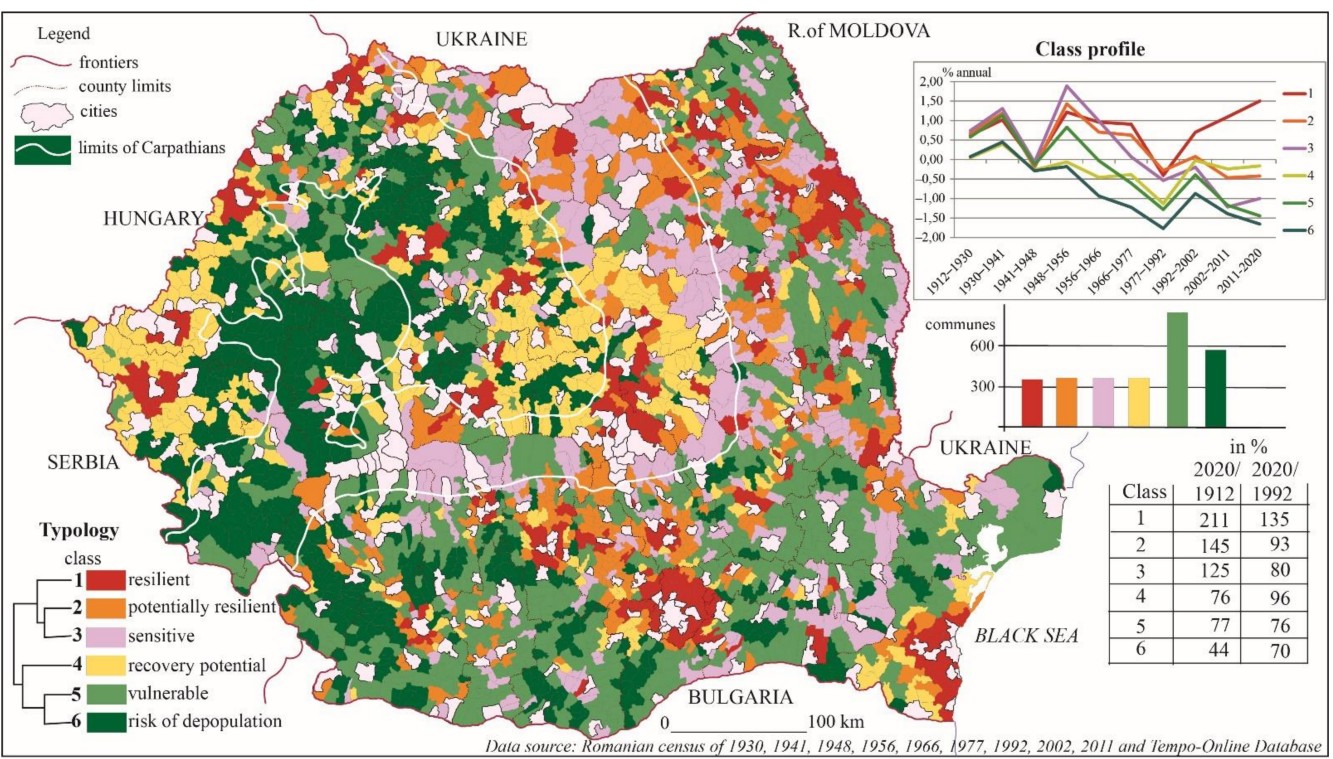

**Figure 1.** Typology of the numerical evolution of the population (1912–2020).

Class 2 experienced a similar evolution to the previous one until 1992 but, subsequently, it registered a slight decline, stabilized around −0.5% annually, largely imputable to its natural deficit. This class can be considered as potentially resilient. Located in the continuity of class 1 or filling in the interstitial spaces in the more densely populated areas (Central Greater Wallachia, north-western Moldavia), the communes belonging to this class can benefit from their favorable position to ensure their demographic stability.

Class 3 groups compact regions in the Carpathian area, as well as certain spaces situated along important rivers (such as the Danube, the Prut and the Siret). The evolution profile, identical to that of the two previous classes until 1948, stands out through the spectacular increase of the population in the first part of the communist period (1948–1966). This can be explained by the intensive exploitation of natural resources (ores and wood in the Carpathian area) or by the development of industrial agriculture in low-lying regions, both situations generating a strong attractiveness. Subsequently, however, the crisis in the mining sector and the modernization of agriculture led to a significant decline, which seems to have worsened in recent decades. This sensitivity has reduced or even eliminated their resilience capacity, the possibility of recovery being, for the time being, minimal.

Class 4 is distinguished by its compact expansion especially in two areas: the plains in the west of the country, on the border with Serbia and Hungary and south-eastern Transylvania, with a diverse ethnic structure (Romanians, Hungarians and Germans) and an early manifestation of demographic transition. In an isolated manner, it can also be encountered in the rest of the country, without forming coherent areas. Although it faced an early decline, accentuated in places towards the end of the communist period by the emigration of the German minority, it has stabilized in recent decades. On the one hand, this recovery potential is due to its favorable position in relation to the European transport system and to the development of agro-industry (in the west of the country). On the other

hand, its recovery potential was induced by the soar of agrotourism and organic agriculture (in south-eastern Transylvania) [54,55].

Class 5 is the most representative, with a remarkable continuity in the south and east of the country, but also including smaller areas in north-eastern Transylvania. Characterized by vulnerability to the risk of depopulation, this category preserved a positive growth rate until 1966, to later enter a decline that has become deeper in recent decades. Characterized by the predominance of agricultural activities, a relative isolation in most cases and the advanced degradation of the population structure by age groups, these communes will most certainly face depopulation, many small villages being already completely depopulated.

Class 6, with a remarkable representativeness, forms an arc of a circle from northern Transylvania to the south of the country, covering most of the Western Carpathians. The risk of depopulation here is not imminent but present, many communes currently having an extremely small population of several hundred inhabitants. This massive decline (on average, the current population is only 44% of that in 1912) is explained by the precocity of the rural exodus from these isolated areas, often adjacent to areas which used to be heavily urbanized and industrialized in the communist period.

The typology thus captures a series of oppositions often previously highlighted in similar studies (peri-urban—deeply rural, mountain-lowland, agricultural—non-agricultural), expressed especially in the abandonment of less productive agricultural lands [56]. The degree to which they influenced the underlined trends was tested by the series of multivariate analyses.

### 4.2. Multivariate Factor Analysis

The main results of the multivariate analysis are shown in Table 3. AGR (average growth rate) is the dependent variable, while the other 16 work as explanatory variables. The R2 coefficient of determination was additionally included, indicating (just like the confidence interval and tolerance), an acceptable level of predictability of the proposed model, both at the national and regional level.

**Table 3.** Correlations between the growth rate of the rural population and the explanatory variables, according to the multiple linear regression, by large historical regions (P1 = 1912–1948; P2 = 1948–1992; P3 = 1992–2020, Confidence interval = 95%, Tolerance = 0.001).

| Variable | ROMANIA | | | Moldavia | | | Wallachia | | | Transylvania−Banat | | |
|---|---|---|---|---|---|---|---|---|---|---|---|---|
| | $P_1$ | $P_2$ | $P_3$ | $P_1$ | $P_2$ | $P_3$ | $P_1$ | $P_2$ | $P_3$ | $P_1$ | $P_2$ | $P_3$ |
| AGR | 1 | 1 | 1 | 1 | 1 | 1 | 1 | 1 | 1 | 1 | 1 | 1 |
| DMC | 0.13 | 0.35 | 0.55 | 0.09 | 0.33 | 0.43 | 0.18 | 0.40 | 0.59 | 0.12 | 0.30 | 0.57 |
| ATI | | | 0.39 | | | 0.29 | | | 0.48 | | | 0.37 |
| ALT | 0.37 | −0.05 | 0.08 | 0.34 | −0.17 | 0.15 | 0.34 | −0.17 | 0.09 | −0.09 | −0.06 | 0.27 |
| DRG | −0.24 | 0.05 | 0.03 | −0.25 | 0.26 | −0.05 | −0.18 | 0.20 | 0.08 | 0.05 | −0.03 | −0.02 |
| AFR | 0.26 | −0.16 | 0.09 | 0.28 | −0.27 | 0.03 | 0.23 | −0.19 | 0.03 | 0.10 | −0.17 | 0.26 |
| SQ | 0.42 | 0.06 | 0.16 | 0.35 | −0.11 | 0.08 | 0.44 | −0.07 | 0.11 | 0.08 | 0.11 | 0.41 |
| HF | 0.06 | 0.32 | 0.33 | −0.15 | 0.34 | 0.16 | 0.27 | 0.20 | 0.33 | −0.06 | 0.46 | 0.48 |
| DNS | −0.12 | 0.20 | 0.39 | −0.42 | 0.02 | 0.31 | −0.20 | 0.24 | 0.49 | −0.03 | 0.11 | 0.45 |
| NB | 0.69 | 0.22 | 0.53 | 0.44 | 0.09 | 0.51 | 0.54 | 0.11 | 0.55 | 0.67 | 0.17 | 0.51 |
| AG | 0.61 | 0.44 | 0.60 | 0.38 | 0.43 | 0.51 | 0.42 | 0.41 | 0.64 | 0.50 | 0.38 | 0.57 |
| PST | | 0.45 | 0.32 | | 0.51 | 0.19 | | 0.61 | 0.46 | | 0.48 | 0.34 |
| MB | | 0.58 | 0.60 | | 0.65 | 0.59 | | 0.76 | 0.63 | | 0.59 | 0.67 |
| HSE | | | 0.25 | | | 0.26 | | | 0.38 | | | 0.29 |
| INC | | | 0.21 | | | 0.08 | | | 0.41 | | | 0.21 |
| NBH | | | 0.48 | | | 0.41 | | | 0.56 | | | 0.47 |
| SWC | | | 0.43 | | | 0.29 | | | 0.61 | | | 0.44 |
| $R^2$ | 0.465 | 0.609 | 0.629 | 0.345 | 0.520 | 0.485 | 0.319 | 0.663 | 0.687 | 0.440 | 0.627 | 0.651 |

A first observation concerns the significant differences between the three periods. Between 1912 and 1948, the quality of the model is slightly lower, in the absence of data on certain variables. The most obvious correlation characterizes the demographic factors (NB and AG),

but the physico-geographical factors (ALT, AFR and SQ) also stand out through significant values. It can be stated that these correlations express the lower level of development of the Romanian rural environment, during this period dependent on the agricultural potential and on the demographic transition, which is an early phase. It is interesting that this dependence is primarily specific to the eastern and southern regions, where the agricultural colonization supported by successive agrarian reforms (1923, 1945) continued until after World War II. This colonization took place mainly in the low, steppe and dry regions of Greater Wallachia, Dobruja and Moldavia, as revealed by the DRG variable. At the same time, the AFR variable also indicates, especially in the case of Moldavia, the expansion of the settlement system towards the forested regions of the Eastern Carpathians, due to the development of logging activities. By contrast, in the case of the Trans-Carpathian regions, the demographic factor was the only significantly correlated one, the altitude or the degree of afforestation being less important, in the context of an earlier and denser population of the low mountainous regions of the Western Carpathians.

The second period, 1948–1992, marked by the communist regime, is very clearly distinguished from the previous one, standing out through the very little significance of physico-geographical factors and the increased importance of some variables that indicate the beginning of the rural exodus, as well as through the rising prominence of accessibility to main cities (DMC) and settlement size (HF). The PST and MB variables, newly introduced in the model, now have the highest incidence, the share of non-agricultural activities and the level of the migration balance becoming more efficient predictors than the natural increase. The AG variable, which maintains its importance (at a lower level however) forms, together with the two previous ones, the main factorial axis for this period, including at regional level. The change of the direction of the correlation with the DRG variable seems to be interesting in the case of Moldavia, a region more clearly marked by the aridity gradient that separates mountainous regions from steppe ones. Compared to the previous period, DRG and AFR basically appear with inverted values in this region, pointing out a superior resistance of the mountain regions in front of the rural exodus. In this sense, a significant role was played by the process of agriculture cooperativization, which massively affected the lowlands, while the mountainous ones almost completely preserved the economic structures inherited from the previous period. For this period, the greater importance of the DMC variable in Greater Wallachia could have been brought about by the presence of the capital city, which was already experiencing the process of periurbanization at that time.

In the case of P3, we can speak of certain continuity in comparison to the previous period, but also of an increase in regional differences. At national level, the recovery of the demographic variables is significant, in the new context of the completion of the demographic transition, which attests to a stronger resilience in the case of the communes with a younger population, less marked by natural deficit. The increased importance of the position in relation to urban centers emphasizes, together with the newly introduced variables (ATI, NBH, SWC, HSE), the expansion of the periurbanization process. Population density, neutral in previous periods, is becoming increasingly important, betraying the same tendency of population concentration in the more densely populated areas situated in the vicinity of some major urban centers. The declining importance of the socio-professional structure of the population is also related to this new context, in which many communes that had experienced an incipient industrialization during the communist period were hit hard by the restructuring of the economy, triggered by the transition to a market economy.

There are larger differences between the three regions than in previous periods. Moldavia records lower values of the correlation coefficients, the growth rate of the population being dictated mainly by the preservation of a certain demographic vitality (NB) or by the level of the migration balance (MB), the region being massively affected by labor emigration. In addition to these factors, Greater Wallachia experiences a greater importance of the access to transport infrastructure, but it also registers a high value of the average income variable (INC), which is illustrative of the strong disparities that characterize it

from this point of view. Not coincidentally, indicators such as NBH and SWC are very well correlated here, while PST and HSE are more important than at the national level. The part played by the concentration of economic activities and population in Central Greater Wallachia, the most densely populated region of the country, is very obvious. In Transylvania-Banat, the correlation between ALT, AFR and QS with the growth rate of the population is interesting, indicating a tendency of population concentration in the lower plain areas, also captured in the typological analysis. The altitudinal contrast here is also related to the precocity of the beginning of the decline, to the more intense humanization of the low mountain massifs, especially in the Western Carpathians. This region is also characterized by the significant correlation between almost all the variables taken into account, even if at an average level, in general. The importance of the position in relation to the urban centers also indicates an advance of the concentration of the population in periurban areas. However, the generalized correlation could also be interpreted in terms of a higher recovery capacity or the existence of a certain resilience potential, as demonstrated by the typological analysis. The more favorable position in the context of the European transport system can be regarded as decisive from this point of view.

To further test the importance of the physico-geographical context, a series of linear regressions was performed at the level of four major types of regions: mountainous (Carpathian), Subcarpathian, hilly and plain. The result (Table 4) reveals their strong specificity and validates the analysis model used.

**Table 4.** Correlations between the growth rate of the rural population and the explanatory variables, according to the multiple linear regression, by large physico-geographical regions (P1 = 1912–1948; P2 = 1948–1992; P3 = 1992–2020, Confidence interval = 95%, Tolerance = 0.001).

| Variable | Carpathian | | | Subcarpathian | | | Hills and Plateaus | | | Plains | | |
|---|---|---|---|---|---|---|---|---|---|---|---|---|
| | $P_1$ | $P_2$ | $P_3$ | $P_1$ | $P_2$ | $P_3$ | $P_1$ | $P_2$ | $P_3$ | $P_1$ | $P_2$ | $P_3$ |
| AGR | 1 | 1 | 1 | 1 | 1 | 1 | 1 | 1 | 1 | 1 | 1 | 1 |
| DMC | 0.09 | 0.15 | 0.49 | 0.25 | 0.39 | 0.54 | 0.12 | 0.36 | 0.51 | 0.01 | 0.48 | 0.61 |
| ATI | | | 0.35 | | | 0.48 | | | 0.35 | | | 0.46 |
| ALT | −0.10 | −0.05 | −0.04 | 0.08 | 0.02 | 0.26 | 0.34 | 0.15 | 0.05 | 0.36 | 0.01 | −0.13 |
| DRG | 0.13 | −0.05 | −0.04 | −0.08 | 0.13 | −0.04 | −0.24 | −0.06 | −0.04 | −0.18 | 0.04 | 0.26 |
| AFR | −0.06 | −0.14 | 0.10 | 0.06 | −0.14 | −0.03 | 0.21 | −0.05 | 0.03 | 0.03 | −0.08 | −0.04 |
| SQ | 0.15 | 0.16 | 0.30 | 0.22 | 0.17 | −0.03 | 0.43 | 0.21 | 0.22 | 0.25 | 0.03 | −0.04 |
| HF | 0.07 | 0.48 | 0.49 | 0.19 | 0.34 | 0.27 | −0.08 | 0.40 | 0.22 | −0.11 | 0.26 | 0.26 |
| DNS | −0.23 | 0.05 | 0.40 | −0.10 | 0.14 | 0.35 | −0.30 | 0.30 | 0.41 | −0.32 | 0.31 | 0.42 |
| NB | 0.59 | 0.39 | 0.56 | 0.42 | 0.20 | 0.54 | 0.64 | 0.24 | 0.56 | 0.76 | 0.21 | 0.54 |
| AG | 0.46 | 0.51 | 0.56 | 0.23 | 0.54 | 0.61 | 0.53 | 0.44 | 0.62 | 0.70 | 0.42 | 0.69 |
| PST | | 0.29 | 0.31 | | 0.49 | 0.26 | | 0.35 | 0.23 | | 0.60 | 0.23 |
| MB | | 0.47 | 0.58 | | 0.61 | 0.62 | | 0.54 | 0.56 | | 0.64 | 0.70 |
| HSE | | | 0.12 | | | 0.23 | | | 0.16 | | | 0.52 |
| INC | | | −0.02 | | | 0.17 | | | 0.05 | | | 0.54 |
| NBH | | | 0.21 | | | 0.37 | | | 0.52 | | | 0.55 |
| SWC | | | 0.34 | | | 0.39 | | | 0.36 | | | 0.71 |
| $R^2$ | 0.387 | 0.639 | 0.793 | 0.219 | 0.643 | 0.799 | 0.475 | 0.652 | 0.792 | 0.603 | 0.685 | 0.862 |

The Carpathian mountain area, forming an arc of a circle that includes three differently oriented chains, stands out through a higher incidence of the factors connected to the position in relation to urban centers, explainable in the context of its relief fragmentation. In counterbalance, habitat fragmentation is much more important than in other types of relief, small localities being disadvantaged in the mountain context. This is a situation which can also be found in other mountainous regions of Europe, including in neighboring states [57]. Another peculiarity is the constant importance of the demographic variables (NB, AG) and the contrast of the incidence of the DNS variable, which from a negative correlation in the first period changed to a consistently positive one in the third period. This situation reflects the expansion of the population system until the second half of the

twentieth century, in more isolated and less exploited mountain areas, by opening some capacities for the extraction of subsoil resources or for the capitalization of the wood mass. After 1990, they fully suffered a setback, which forced the population to concentrate in more densely populated areas, usually in depressions and wider valleys [58].

The Subcarpathians group the regions of contact to lower areas, south and east of the Carpathians, densely populated and traditionally urbanized. Unlike the neighboring Carpathian area, the distance to important urban centers is well correlated and, towards the end, altitude imposes itself, revealing a tendency to concentrate in lower areas, either of depressions or contact with plain regions. In the first part of the 19th century, soil fertility was still significantly correlated, its importance being later overshadowed by the fragmentation of the habitat, in keeping with the general tendency of the population to concentrate in larger settlements. This explains the important correlation with the NBH and SWC variables, the urbanistic level being dependent on the settlement size [59].

The hilly and plateau areas cover most of the Romanian territory, in continuity to the east and south with the Subcarpathians or occupying the space inside the Carpathian arc. As everywhere, the role of the city has gradually increased, together with the dependence on the transport and urbanistic infrastructure or with the diversification of economic activities. A peculiarity is the constant importance of soil quality and the strong contrast of the incidence of habitat fragmentation and population density. The hilly areas, just like the Subcarpathian ones, are the most affected by geomorphological processes, especially in those areas with a strong relief fragmentation, generating a network of predominantly small settlements, vulnerable to the effects of the rural exodus. In the case of the last period, the strong correlation of the NBH variable indicates a stronger contrast between the settlements favored by their proximity to urban centers and the isolated ones, subject to a chronic decline.

The plains, which occupy a peripheral position in the west and south of the country, best express the population's tendency to concentrate around major cities through the very high values of the DMC, ATI, HSE, INC, NBH and SWC variables. The very high value of the R2 determination index shows that the model used is very well adapted in the case of this type of regions. For the last period, the significant correlation with the DRG variable is also interesting, the drier plain regions in south-eastern Romania being much more vulnerable to the depopulation process, unlike the plains in the west of the country, correlated with the global climate changes that induce aridity trends, as certified by some studies [60].

In the cross-sectional profile, it is interesting that some variables record an increase in their correlation index, both in time and space, in an ascending direction. For example, the importance of the position in relation to important urban centers is rising from mountainous to plain regions, along with the population density, the aging index, the migration balance and the variables introduced for the last period. All indicate a higher contrast in the plain regions, in relation to the access to the infrastructure and services that primarily favor developing metropolitan areas.

## 5. Conclusions

The presentation and interpretation of the results of the analyses performed according to the exposed methodology validate the hypotheses that formed the basis of the study. The limitations induced by the absence of information on certain variables and periods did not diminish the applicability of the model to the analysis scales used.

The typological analysis certified the importance of investigating the numerical evolution of the population as an indicator of the two competing processes, rural decline and periurban concentration. The trends extracted from the analysis of the profile of the evolution types quite clearly indicate the capacity for demographic resilience, especially when long data series are used. The regionalized distribution of types and their spatial coherence demonstrates the importance of such studies for the territorial diagnosis of the potential for sustainable development. The greater (active or potential) capacity for demographic

resilience in the areas situated in the vicinity of major cities is consistent with the modern trends of metropolitan population concentration. The recovery capacity of certain regions (plains in the west of the country, south-eastern Transylvania) is very well captured by the typological analysis, emphasizing the importance of the geographical position or the capitalization of sustainable development opportunities (tourism, organic farming), in line with the rural development strategy of the country for the period 2014–2020 [61]. A significant result is the highlighting of the six types of specific situations, which require well-oriented strategies, aimed at stopping regressive trends and focusing on a possible recovery, even in the context of a lower population. Each of these situations requires tailored measures, predominantly towards sustainable resource use in mountain areas or towards making peasant agriculture in the hilly and plain areas more efficient. Official documents, such as the one mentioned above, stipulate sets of measures granted with European directives, but only at a general level, being insufficiently anchored in the local reality, although there are strategies developed at the county or local level.

Factor analyses, both on the national and regional scale (cultural-historical or physico-geographical) emphasized the specific incidence of each factor. Thus, the tested model could be replicated on a more detailed scale in order to properly guide local development strategies. Complementary to the typological analysis, the results obtained from multiple regressions certified the essential role of some variables related to accessibility (position in relation to urban centers, access to transport infrastructure, diversification of economic activities and services). The physico-geographical variables are equally important, even if their incidence had a sequential manifestation in time and space. The altitude-related vulnerability (especially in mountainous areas) and the drought-related one (in the hilly and plain areas in the east of the country) deserve special attention and can serve as a subject for more applied studies. The very strong correlation between the growth rate of the population and the urbanistic level also represents a significant result, in the context of the precarious quality of life in rural Romania. Drawing up firm measures meant to reduce the gaps in this direction becomes imperative in order to temper depopulation trends in the short and medium term, trends which are more than preoccupying in the fragile areas spotlighted by the typological analysis (types 5 and 6). The objectives advanced by rural development strategies, on various scales of analysis, have rarely been achieved, the progress being more palpable, for example, in terms of improving road infrastructure but much weaker in terms of expanding utility networks or stimulating some forms of local cooperation.

The present study could serve as a basis for current demographic resilience analyses by introducing more specific variables in the analysis model in order to spot the vulnerable dynamic or structural aspects, beyond the overall picture provided. A prospective, medium-term study is also needed to capture the extent of the changes undergone by demographic territorial structures in the event of maintaining current evolution trends. These investigations can contribute to the implementation of an administrative-territorial reform, always postponed for obscure reasons, the current structure being created in 1968 to correspond to the desiderata of a centrally planned economy. The need to shape a new territorial development framework, adapted to the current situation, is deeply felt in rural areas, often disconnected from modern economic circuits.

As in much of Eastern Europe, however, rural depopulation seems to be becoming the new normal of the 21st century in Romania. This inevitable process of transition from an agrarian to an urban-industrial economy and, further on, to a knowledge economy, is difficult to stop; it can only be softened, given its structural, complex character. The challenges, risks and associated vulnerabilities cannot be ignored. There is often a lack of a sustainable alternative to managing or adapting to the consequences of depopulation and turning it into an opportunity by creating a model of sustainable and innovative population growth, making the most of the existing potential. In order to overcome external changes, isolated communities, severely affected by depopulation, need to improve their resilience capacity by adjusting their functionality and internal structure. Although hardly on the

political agenda, the issue of depopulation, accompanied by the precariousness of the skilled workforce and the aging population, calls for increased, targeted investment in education and training.

The case of Romania is paradigmatic from the viewpoint of demographic resilience as it illustrates a highly adaptive but also vulnerable system that suffered many radical transformations when transitioning from a predominantly rural system before World War II to a balanced systematized urban–rural system during communism and, finally, to a more chaotic and diverse territorial system influenced by both in-migration (counterurbanization, peri-urbanization) and out-migration (external emigration) processes. It is a case that is at the same time specific for all the former socialist-planned economies but also unique because of the turbulent historic evolution of this territory that was always at the crossroads between divergent and sometimes conflicting political and economic systems. Further studies could analyze more in-depth the phases of demographic resilience in relation to the model of the adaptive cycle and test more comprehensive models of adaptive and evolutionary resilience by integrating more drivers and causal explanations of past and present social-economic context. Meanwhile, the study area could be enlarged to all former communist European states by comparing, for example, those that are presently members of the European Union to the non-member countries.

**Author Contributions:** Conceptualization, I.M. and M.I.; methodology, I.M. and A.B.; validation, I.M. and A.B.; formal analysis, M.I. and R.I.H.-Ș.; writing—original draft preparation, I.M.; writing—review and editing, M.I., A.B. and R.I.H.-Ș.; supervision, I.M. All authors have read and agreed to the published version of the manuscript.

**Funding:** This research received no external funding.

**Informed Consent Statement:** Not applicable.

**Data Availability Statement:** Not applicable.

**Conflicts of Interest:** The authors declare no conflict of interest.

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
