# Peer review of "Demographic Resilience in the Rural Area of Romania. A Statistical-Territorial Approach of the Last Hundred Years"

_sustainability, doi:10.3390/su131910902_

Round 1

Reviewer 1 Report

I sincerely congratulate the authors of the paper, because they focused on a very interesting issue and used an interesting research method. I read the text with great pleasure.

Many countries are currently facing major demographic challenges and the risk of depopulation is one of them. These problems are especially visible in rural areas. That's why I believe, the topic discussed in this paper is very current and important from the social, economic and administrative point of view.

The article contains the appropriate structure. The article has been correctly divided into relevant sections, and their content coincides with their titles.

It seems that it would be good to supplement the paper with some information that is not obvious to readers outside of Romania. I'm referring to a more in-depth description of the causes of the demographic problems and their consequences, especially in rural areas of Romania.

Research hypotheses were clearly defined.  

Footnotes and bibliography are correctly formulated.

 The language of the article is mature, correct, adequate. The paper is aesthetic.

A lot of space and efforts in the paper was put on the research part. The conducted analysis is extensive and correctly performed. I believe it was well designed and made. The applied research methods are adequate to the problem. I have no comments here.

In the context of such a topic, the section on literature review could always be extended.

Author Response

Dear Reviewer,

Thank you for your review.

We have tried to make all the necessary modifications in order to meet your expectation. In the attached file you can find the solutions we have found.

If there are other revisions to be made, we are at your disposal.

Best regards,

The authors of the paper Sustainability 1344762: Demographic resilience in the rural area of Romania. A statistical - territorial approach of the last hundred years

Reviewer 2 Report

First of all, I would like to evaluate positively the relatively comprehensive view of the issue of demographic development of the Romanian countryside. Nevertheless, I have some important and a few minor comments.

In the demographic article, we should condition the development not on changes in regimes, but on demographically relevant changes. It is a transition from pre-modern to modern society, which is accompanied by the first demographic revolution, and later a transition from modern to postmodern society, accompanied by a second demographic transition. However, it is possible that in the Romanian case these transitions may coincide with changes in political regimes, but this should be examined in this study. The opportunity to do so was due to the collection of large amounts of data over 100 years. After all, the authors suggest some of these connections in the following text - we just need to build on that theory. In this sense, the literature should be supplemented. After all, the transition to post-modern society has not yet progressed sufficiently in Romania. In that case, rural-to-urban migration would be complemented by strong urban-to-rural migration and the problem of rural depopulation would be limited to only some particularly disadvantaged regions.

Hypotheses should define the facts that should be confirmed or refuted in the work. To this end, the measurable state to be achieved should be determined as accurately as possible in order to check that the target state has been reached. In our case, the hypotheses do not fulfill this function. I recommend omitting hypotheses and instead clearly defining the goal of the work. Is it a verification of the methodology, an analysis of the demographic development of the Romanian countryside, a proposal for resolving the situation?

It would also be necessary to define what is meant by rural depopulation. European villages have disappeared since the Middle Ages. It is a natural evolution. Originally, when peasants farmed their fields with primitive means and moved on foot, a dense network of rural settlements was needed to get to the fields, work, and return in one day. With the development of production, transport and other technologies, such a dense network of villages is no longer needed and residents are concentrating in larger and better equipped settlements along transport lines. The demise of individual settlements (which usually do not disappear physically, but turn into second-homes residences) is therefore no tragedy - maybe it is a cultural loss. Rather, depopulation risk should be understood as depopulation of entire rural regions, threatening the disintegration of the settlement structure.

I support the authors' thesis that there are important differences in the quality of life for migration trends in rural areas. At the same time, I support a certain skepticism about the possibility of replacing job losses with the development of tourism. This may help resolve the situation, but it is not enough in itself. The lower formal education of the rural population is a natural characteristic of the countryside, as prestigious and well-paid job opportunities are concentrated in large cities. However, not all villagers have these ambitions, and moreover, educated people have recently begun to move to the countryside in the context of suburbanization, distance employment and as seniors.

As for Class 1: these municipalities are perhaps saved in terms of current demographic development, but they have often lost their rural character (or part of it) and become dependent on the city in question. At the same time, they have a number of problems in common with housing estates on the outskirts of large cities, and in the not entirely distant future, ghettos of certain social groups may become involved. If the increase in population is the result of one wave of development, an abnormal one-sided demographic and social structure arises, which creates possible problems for the future. Therefore, I would be careful with a too positive evaluation of municipalities of the Class 1.

As Romania has been a significant country of emigration for the last 30 years, it would be good to at least mention the impact of foreign migration. Is there a rural-to-urban-to-abroad migration model or do rural people emigrate directly abroad? What is possible to expect in the future?

For Table 1, it would be necessary to define what the authors consider to be urban and what rural. In keywords, I recommend replacing the word drivers with the word driving forces. I recommend replacing the word Chinese P.R. (line 90) with P. R. China (maybe China is enough). The problem of underestimating the concept of sustainable rural development may lie in the idea that sustainability is primarily associated with the environment and that rural settlements are not endangered in this respect (which may not be true). On the line 309 Romanians or Roma is meant? On the lines 317 and 324 new paragraphs for Classes 5 and 6?

In my opinion, it would first be necessary to somewhat rework the theory and supplement the relevant literature.

Author Response

Dear Reviewer,

Thank you for your review.

We have tried to make all the necessary modifications in order to meet your expectations. In the attached file you can find the solutions we have found.

If there are other revisions to be made, we are at your disposal.

Best regards,

The authors of the paper: Sustainability 1344762: Demographic resilience in the rural area of Romania. A statistical-territorial approach of the last hundred years

Reviewer 3 Report

I like this paper and I am in favor of publication. However, I see moderate/major revisions necessary to improve the quality prior to acceptance.

First, the language usage is likely the most relevant problem. Some phrases are not understandable in the present form. Think about the abstract: "as a support for multivariate analysis"... Would you mean "with the support of multivariate analysis"? Please check the whole text extensively.

Second, the definition of demographic resilience is not totally clear to me. It is poorly generalizable and lacks normative and applied references for contextualization.

Third, a discussion on the quality of official statistics is necessary and actually lacks in the manuscript.

Why Romania is a paradigmatic case study for demographic resilience?

The urban-rural gradient in Romania is a protagonist of demographic resilience. Why don't refer to the approach of a recent paper from Serra P. et al. (2014) Applied Geography, for better contextualize this point?

Can you delineate future research steps in Romania and at a broader scale?

Author Response

(The authors gave the same response as above.)

Round 2

Reviewer 3 Report

Good revisions overall. I like the paper. I believe it is publishable in the present form with customary language checks performed by MDPI.